# Population-Based Study on Cancer Incidence in Pharmacist: A Cohort Study in Taiwan

**DOI:** 10.3390/ijerph182312625

**Published:** 2021-11-30

**Authors:** Ya-Wen Lin, Che-Huei Lin, Lee-Wen Pai, Chih-Hsin Mou, Jong-Yi Wang, Ming-Hung Lin

**Affiliations:** 1School of Nursing, China Medical University, Taichung 406040, Taiwan; wen5001@yahoo.com.tw; 2Department of Public Health, China Medical University, Taichung 406040, Taiwan; 3Department of College Business Administration, National Chiayi University, Chiayi 600355, Taiwan; fulunsun@yahoo.com.tw; 4College of Pharmacy and Science, Chia Nan University of Pharmacy & Science, Tainan 71710, Taiwan; 5Department of Nursing, Central Taiwan University of Science and Technology, Taichung 406053, Taiwan; lwpai@ctust.edu.tw; 6Management Office for Health Data, China Medical University Hospital, Taichung 40447, Taiwan; b8507006@gmail.com; 7Department of Health Services Administration, China Medical University, Taichung 406040, Taiwan; 8Department of Nursing, National Taichung University of Science and Technology, Taichung 40343, Taiwan

**Keywords:** pharmacists, occupational, cancer, risk factor

## Abstract

Cancer is increasing in rate globally and is leading cause of death among no communicable chronic diseases (NCDs) after cardiovascular disease (CVD). Most of the research focuses on the risk of occupational injury, job stress, mental illness, substance abuse and workplace safety in physicians and nurses. However, fewer studies have investigated the risk of cancer in pharmacists. We compared the matched general population to investigate the risk of cancer in pharmacists in Taiwan. Data were obtained from the Health and Welfare Data Science Center of the Ministry of Health and Welfare in Taiwan. We established a pharmacist group that included 11,568 pharmacists and selected a 4-fold comparison (*n =* 46,272) for the non-clinics comparison group, frequency being matched by age, gender and Charlson Comorbidity Index (CCI) score. The pharmacists had a lower but non-significant risk of all cancer (Adjusted hazard ratio [aHR] = 0.96; 95% confidence interval [CI] = 0.85–1.07) compared with the general population. Female pharmacists had a higher risk of cancer than male pharmacists ([aHR] = 1.23; 95% CI = 1.06–1.43). Pharmacists had higher risks of breast cancer in females (aHR = 1.68; 95% CI = 1.35–2.08) and of prostate cancer in males (aHR = 2.18; 95% CI = 1.35–2.08) when compared with the general population. Occupational risk factors could play a role, but they were not evaluated. These epidemiological findings require additional studies to clarify cancer risk mechanisms in pharmacists.

## 1. Introduction

The health sector has been termed as one of the most hazardous environments to work in, as employees, such as those working in hospitals and health facilities, are constantly exposed to a complex variety of occupational risks in the course of their work [1]. Occupational risks in clinical and non-clinical practitioners can vary depending on their specific profession, the nature of their work and their workplace type and environment [2]. As highlighted by Che Huei et al. [3] and Lombardo and Roussel [4], employees in hospital settings are exposed to physical hazards such as radiation; psychological hazards such as shift work; biological hazards such as bacterial or viral infections; and chemical hazards such as antineoplastic drugs.

Research has directly linked several types of cancers to occupational hazards, namely carcinogens (Occupational Safety and Health Administration, Ministry of Labor) [5], the most common types of occupational cancer being lung cancer, bladder cancer and mesothelioma [6]. Estimates of the recent and future burden of occupational diseases indicate that occupational cancer is a pervasive problem owing to the exposure of workers to carcinogenic agents [7]. Some studies define carcinogens as any substance, reagent, radionuclide or radiation that can cause cancer in humans through the carcinogenic process [8,9]. The National Institute for Occupational Safety and Health (NIOSH) estimates that around 8 million HCWs need to manage hazardous drugs in their workplace. Although work practices and safe hazardous drug handling practices have improved, HCW’s exposure to hazardous drug remains a problem. Pharmacists need to prepare antineoplastic agents as part of their work practice, and they are more likely to be exposed to hazardous drugs than the general population. They are more likely to develop certain types of cancers than the general population [10].

Cancer, also known as neoplasia or malignancy, is a broad term used to refer to a large collection of more than one hundred diseases (cancers) that result in the abnormal proliferation of any of the different kinds of cells found in various organs or tissues of the human body [11]. As stated by Khan and Farhana [12], the fundamental abnormality in cellular function that distinguishes cancer cells from normal cells is their incessant and unregulated proliferation. Tumors can be cancerous or non-cancerous (benign). Cancerous tumors are also called malignant tumors. Cancerous tumors can multiply and spread to nearby tissues form new tumors through metastasis, thereby increasing the risk of cancer morbidity and mortality [11].

Cancer cases have been increasing, and it is the second leading cause of death globally, accounting for nearly 10 million death in 2020 (World Health Organization, 2020) [13]. Cancer causes tremendous physical, emotional and economic pressure on individuals, families, communities and health system [14]. According to World Health Organization data, the estimated number of prevalent cases (1-year) as a proportion (per 100,000) of ages 20+ is 59.7 for breast cancer in females and 16.2 for prostate cancer in males in China. The estimated age-standardized incidence rate of breast cancer in females is 65.2, and that of prostate cancer of males is 17.1. It is estimated that from 2020 to 2040, there will be an increase of 25.8 million new cases of breast cancer in women and 27 million new cases of prostate cancer in men over the age of 20 in Asia (International Agency for Research on Cancer, [IARC] 2020) [15].

It is estimated that 1 in 5 people are diagnosed with cancer during their lifetime, and the disease kills 1 in 8 men and 1 in 11 women (Union for International Cancer Control [UICC], 2021) [16]. The number of prevalent global cases within a 5-year period is 50 million people (IARC, 2020) [15]. The economic impact of cancer is significant and is increasing. According to estimates, the annual economic cost of cancer is approximately 1.16 trillion USD [17]. The most common cancers in the world are lung, breast, colorectal, prostate, skin (non-melanoma) and stomach cancer, in that particular order, while the most common cancer deaths result from lung, colorectal, stomach, liver and breast cancer, in that particular order [14,17].

In Taiwan, the rankings for cancer incidence, morbidity and mortality are high. Cancer is a leading cause of death in the country [18]. The cancer incidence rates for males and females were 485.1 and 414.4 per 100,000 people, respectively, according to 2016 cancer registry data [18,19,20]. The age-standardized incidence rates for males and females were 330.0 and 269.1 people per 100,000 people, respectively. The leading incidence rate (per 100,000 people) of a cancer site was colon in male and breast in female (49.3 and 73.0, respectively). The standardized cancer mortality rate in 2018 was 121.8 per 100,000 people. The mortality rate increased from 169.2 to 206.9 from 2008 to 2018. The leading causes of cancer deaths are the trachea, bronchi and lungs, followed by the liver and intrahepatic bile ducts, while the breast (female) ranks fourth (39.8, 34.9 and 20.4 per 100,000 people) [18,19,20].

The total case load continues to rise as new cases are diagnosed. According to the most recent cancer data update conducted in 2018, a total of 116,131 people are diagnosed with various types of cancer, an equivalent of one new case every four minutes and 31 s (Health Promotion Administration [HPA], Ministry of Health and Welfare [MOHW], 2020) [19,21]. The most common cancers in the country are colorectal, lung, breast and liver cancer. Other significant types are oral, prostate, thyroid, skin, stomach and endometrial cancer, as well as liver, thyroid, colon rectum, kidney, nasopharynx and bladder cancer [18,19].

The incidence of cancer has increased in Taiwan; the age-standardized incidence rates (ASIR) for men and women were 341.30 and 284.66 in 2018 per 100,000 people, which was 1.04-fold higher than that in 2014. The leading cancer site in terms of incidence rate is breast cancer, the second is colon rectum cancer, and the fourth is prostate cancer [19,21]. In terms of diagnoses by gender, males account for 61,779 of the new cases (341.3 new cases per 100,000 males) while females accounted for 54,352 of the cases (284.7 new cases per 100,000 females) [19,21]. The most common types of cancers in men are colorectal, lung and liver cancer, in that particular order, while in women, the most common cancers are breast, colorectal and lung cancer, in that particular order [19,20]. The rise in incidence, morbidity and mortality of cancer has led to a steady rise in the clinical use and costs of all anticancer drugs and other treatments including chemotherapy. The costs of novel therapeutics and immunotherapies account for nearly two-thirds of the total antineoplastic agent usage in Taiwan [18].

### The Present Study

Pharmacists and the general population in hospitals and other healthcare facilities constitute a substantial share of the occupational sector in Taiwan’s working population (Ministry of Health and Welfare [MOHW], 2020) [20]. Recent data indicates that there are more than 312,887 HCWs, including 34,838 pharmacists and assistants, working in the health and medical care sector in Taiwan [20]. There are also more than 20,000 non-clinical staff working in hospitals and clinics in Taiwan (National Development Council [NDC], 2016) [22]. All these workers are, by the nature of their work and workplace environment, as well as other risk factors such as gender, age, workplace and work years, exposed to various potential carcinogens [4]. While many studies [23,24,25,26] have focused on cancer risk in physicians, doctors and nurse, few studies have investigated the risk of cancer in pharmacists. Lnug cancer, liver cancer, colorectal cancer, breast cancer and nasopharynx cancer were the leading causes of cancer-related deaths in Taiwan. Compared with the general population, pharmacists are more likely to be exposed to dangerous drugs and develop certain types of cancer. In this population-based study, we were focused on investigating the risk and incidence of the most common cancers among Taiwanese pharmacists.

## 2. Methods

### 2.1. Study Design

The present study was a nationwide population-based cohort study: the study followed up and observed longitudinally pharmacists and general population to assess for primary cancer. All subjects were followed up from 1 January 2000, and the end point (primary cancer development) was the end of 2011 [9,27,28].

### 2.2. Data Collection

#### 2.2.1. Data Sources

Data were obtained from National Health Insurance (NHI) program released by the Ministry of Health and Welfare (MOHW) in Taiwan. The data are nationwide population-based data that follow up patient activity in the country’s healthcare system for 12 years and therefore are an excellent resource for assessing cancer risk in pharmacists [29]. Instituted in 1995, the NHI program is mandated to offer compulsory and comprehensive inpatient and outpatient health and medical care insurance coverage for illnesses, injuries, and childbirth benefits to all citizens and legal residents in the country, except prison inmates. The NHI program has to date achieved nearly 99% enrollment among Taiwan’s population of more than 23.3 million (National Health Insurance [NHI], n.d.) [30].

The data include personal identification numbers, sociodemographic information and medical history including diagnoses, procedures and prescribed medications. To ensure the privacy of the participants, all the data were linked with surrogate identifications processed by the NHRI before being released to researchers. The National Health Insurance Research Database (NHIRD) encrypts patient personal information to protect privacy and provides researchers with anonymous identification numbers associated with relevant claims information, including sex, date of birth, medical services received and prescriptions. Therefore, patient consent is not required to access the NHIRD. This study was approved to fulfill the condition for exemption by the Institutional Review Board (IRB) of China Medical University and Hospital (CMUH-104-REC2-115-CR4). The IRB also specifically waived the consent requirement.

The database provided by the ministry also contains information on medical personnel, including pharmacists and other healthcare providers. Available information includes specialty, date licensed, work area, hospital level, types of employment and claims data (NHIRD, n.d.) [31,32].

#### 2.2.2. Outcome Measures and Risk Factor

In this study, we aimed to investigate the incidence of primary cancer among Taiwanese pharmacists. For each patient, the identification of cancer diagnosis was based on the International Classification of Diseases, Ninth Revision, Clinical Modification from the catastrophic illness registry. Types of cancer were categorized as: Liver (ICD-9 code: 155), Breast (ICD-9 code: 174), Lung (ICD-9 code: 162), Thyroid (ICD-9 code: 193), Colon rectum (ICD-9 code: 153–154), Prostate (ICD-9 code: 185), Kidney (ICD-9 code: 189), Nasopharynx (ICD-9 code: 147), Stomach (ICD-9 code: 151), Bladder (ICD-9 code: 188) and other cancers (ICD-9 codes: 140–146, 148–150, 152, 156–161, 163–165, 170–173, 175–176, 179–184, 186, 187, 190–192 and 194–208). The risk factor included demographic characteristics and Carlson’s comorbidity index (CCI) score. The demographic characteristics included: age; gender (sex); work years (number of years worked as a pharmacist); and workplace (medical center, regional hospital, district Hospital, clinic and pharmacy).

#### 2.2.3. Study Population

The study identified 12,920 pharmacists from 22,221,706 Taiwanese citizens. The exclusion criteria were: (1) without license before 2000 (*n* = 1282); (2) without practicing (*n* = 5); (3) with cancer history (*n* = 63); and (4) with age >100 or <20 years, (*n* = 2). The remaining 11,568 were analyzed. In the non-clinics (*N* = 21907670), the exclusion criteria were: (1) with cancer history (*n* = 162,802); and (2) with age >100 or <20 years (*n* = 6,542,772).

The comparison cohort was randomly selected from the non-clinic people by frequency matching for age, gender and CCI score, with a matched ratio of 4:1, which means that the 4-general population were selected for each case of pharmacists. Figure 1 is the flowchart showing the process by which the study population was selected.

#### 2.2.4. Statistical Analyses

The statistical software of SAS, version 9.4 (SAS Institute, Cary, NC, USA) was used in the analysis. The distribution of demographic characteristics was presented by number and percentage for categorical variable, and mean and standard deviation (SD) for continuous variable. Cancer incidence was calculated in both cohorts using the following formula: sum of cancer development divided by sum of follow-up years (person-years). Cancer risk in the pharmacist cohort compared to the comparison cohort was assessed by the Cox proportional hazard model. The adjusted Cox model was controlled by age, gender and CCI-score. Age, gender and CCI score stratified cancer risk were also estimated. The interaction of cancer risk between pharmacists with demographic characteristics was by the Cox model. Different locations of cancer risk were estimated by gender. The association between cancer risk and the study cohort’s work year and workplace was also assessed. Significance was set at *p* < 0.05 (2-tailed).

## 3. Results

### 3.1. Demographics Characteristics of Study Groups

A total sample of 57,840 (general population, *n* = 46,272; pharmacists, *n* = 11,568) without cancer history at index date (1 January 2000) were included in the analysis. Table 1 shows the general characteristics and demographics of the participants, including age, gender and CCI score. The mean age of the pharmacists (study cohort) was 37.4 years (standard deviation [SD] = 9.50), while that of general population was also 37.4 years (SD = 9.51). The majority (46.7%) of the study cohort were 20–34 year olds, while the least (46.7%) were 55+ year olds. In between were 35–44 year olds (30.9%) and 45–54 year olds (17.8%). Most pharmacists were women (59.1%); had a CCI score of zero (98.7%); worked in clinics (33.4%); and worked for 3–5 years (35.5%).

### 3.2. Comparison of Cancer Risk between Pharmacists and General Population

In total, 3.3% of the pharmacists (383 of 11,568) and 3.3% of the general population (1,547 of 46,272) developed cancer during the follow-up. The cancer incidence rate in pharmacists was 0.95-fold lower than that in the general population (2.83 vs. 2.96 per 1000 person-years) (Table 2). After adjusting for age, gender and CCI score, pharmacists had a lower cancer risk than the general population but did not achieve the statistical significance (aHR = 0.96, 95% CI = 0.85-1.07) (Table 2).

A stratified analysis of cancer risk between pharmacists and general population comparisons was done on gender, age and CCI score level (Table 2). In terms of gender, female pharmacists had a significantly higher cancer risk than the general female population (aHR = 1.23, 95% CI = 1.06–1.43). However, male pharmacists had a significantly lower cancer risk than the general population (aHR = 0.73, 95% CI = 0.61–0.86).

The cancer incidence increased with age in both cohorts. In each age group (in years), pharmacists and comparisons had a similar cancer risk. Cancer incidence was the lowest in the CCI = 0 group for both pharmacists and the general population (Table 2).

In the analysis of individual cancer risk between pharmacists and comparisons (Table 3), pharmacists had a significantly higher risk for prostate cancer in men (aHR = 2.18, 95% CI = 1.12–4.21) and breast cancer in women (aHR = 1.68, 95% CI = 1.35–2.08) but had a significantly lower risk for liver cancer (aHR = 0.50, 95% CI = 0.33–0.74), lung cancer (aHR = 0.61, 95% CI = 0.68–0.98) and other cancers (aHR = 0.69, 95% CI = 0.55–0.86). Except for prostate cancer, male pharmacists had a significantly lower risk of liver (aHR = 0.46, 95% CI = 0.30–0.72) and other cancers (aHR = 0.58, 95% CI = 0.43–0.79).

### 3.3. The Association between Cancer Incidence and Pharmacist Characteristics

The results showed that cancer incidence increased as age and CCI score increased (Table 4). Women had a 1.59-fold cancer risk compared to men (95% CI = 1.27–2.00). Those working in a pharmacy had a 1.59-fold risk compared to those working in a regional hospital (95% CI = 1.07–2.37). In both genders, although those working in a pharmacy still had a higher cancer risk, this did not achieve statistical significance. There were no statistically significant differences between women and men pharmacists in all cancer incidences and risks by work years (*p* > 0.05) and workplace (*p* > 0.05) (Table 4). 

## 4. Discussion

A notable finding in in this study was that there was a significant risk of cancer in employees working in hospitals and healthcare facilities in Taiwan. An important number of employees (both pharmacists and those in the general population, and both women and men) developed cancer within the first 1–2 years of working in medical centers, regional hospitals and clinics. Ten different types of cancer identified during the study period included: Breast, Colon rectum, Liver, Thyroid, Lung, Prostate, Kidney, Nasopharynx, Stomach and Bladder. Cancer was identified in 1930 employees (383 pharmacists and 1547 general population individuals). The incidence of breast cancer is highest, and the incidences of nasopharyngeal, stomach and bladder cancer are the lowest.

An interesting finding was that the cancer risk of pharmacists is lower than that of the general population. One possible explanation for this is that pharmacists, as healthcare workers (HCWs), have better medical and cancer knowledge and awareness, more economic resources and more evidence-based hazard control mechanisms that enabled them to adopt and practice hazard and cancer healthy personal and workplace behaviors to counteract occupational risk mitigation strategies than did the general population factors [33]. Although there is a lower cancer incidence rate in pharmacists in comparison with the general population, HCWs are more stressed because of increasing workload, smaller staffs, longer working hours and the hazards of the workplace. However, because of data limitations, we cannot further analyze the possible risk factors [34].

When an analysis was done based on the gender subgroups, a notable finding was that female pharmacists had higher cancer risk than did those in the general female population. However, male pharmacists had a lower cancer risk than did male non-clinical comparisons. Despite the lower cancer risk, male pharmacists had a significantly higher risk of prostate cancer than those in the general male population. Female pharmacists also had a substantially higher risk and incidence of breast cancer compared to those in the general female population. The higher incidence and risk of breast cancer among female pharmacists may be attributed to occupational factors, and environmental stimuli. Previous epidemiological and experimental studies have confirmed that female pharmacists are susceptible to cancer due to exposure to night work, pesticides, polycyclic aromatic hydrocarbons and metals [35,36,37]. The workplace also provides regular inspections, which may result in a higher cancer detection rate in female pharmacists than in their comparisons [35].

While middle aged pharmacists (35–44 years) were shown to have a statistically nonsignificant higher cancer risk than the general population of the same age group, pharmacists aged 20–34 years old, 45–54 years and 55+ years had a lower cancer risk than did the general population of the same age groups, respectively. This finding contradicts other research and government reports, where it is indicated that nearly all new cancer cases in Taiwan are diagnosed in people aged at least 63 years (median age of diagnosis) [19,20]. The study agreed with others that the cancer incidence rate (IR) is higher in males than in females [19,20].

Pharmacists working in medical centers were found to have a higher cancer risk than the general population, although not significantly so. The cancer risk of pharmacists working in regional hospitals, clinics and pharmacies is expected to be higher than that of the general population working in these institutions, respectively. However, the results of the study showed that pharmacists working in the above-mentioned places (regional hospitals, clinics and pharmacies) had a lower cancer risk than the control group. This study showed that male pharmacists working in a pharmacy have a higher risk of developing cancer than those at other workplaces. There are many chain pharmacies in Taiwan and most of them are male pharmacists in charge. The person in charge of pharmacy must be responsible for the sales, management and operation of the pharmacies himself. Male pharmacists working in pharmacies are under more pressure [38].

The present study showed that the top six cancers in pharmacists in Taiwan were prostate, breast, kidney, thyroid, bladder and nasopharynx cancer, in that particular order. The findings are similar to the ranking of cancer incidence, morbidity and mortality in the general population of Taiwan, where the top four cancers are colorectal, lung, breast and liver cancer [18,19,20].

When age, gender, workplace, location/type of cancer and the CCI score are taken into consideration, the findings of the present study suggested that, despite a lower all-cancer risk, pharmacists are more likely to develop certain types of cancers than the general population, such as breast, thyroid, prostate, nasopharynx and bladder cancer. This finding is also supported the literature [12,19,39]. It is a concern that the workplace, particular medical centers, exposes pharmacists to a higher risk of cancer. It provides useful epidemiological information and inspires the need for future research on the underlying cancer risk factor and mechanisms.

A possible factor and mechanism for the higher rate of prostate and breast cancer in male and female pharmacists is work-related exposure to carcinogens and the practice of pharmacy itself, which includes long-working hours and working on the feet. The present study showed that the workplace is an important factor that exposes employees to several important occupational cancers, an insight that is also supported in the literature [3]. The study showed that pharmacists are constantly exposed to a complex variety of occupational risks in the course of their work, an observation also made in the literature [1]. As demonstrated, and also supported by research, occupational risks in clinical and non-clinical practitioners can vary depending on their specific profession, the very nature of their work and the unit or type of the hospital or healthcare facility in which they are attached [2].

A large body of research and practitioner literature has alluded to intrinsic and extrinsic cancer risk factors: intrinsic factors include age, sex, genetics, hormones, immunity, metabolism and nutritional status; and extrinsic factors include lifestyle (such as certain foods, drinks, cigarette smoking and sexual behaviors) and environmental conditions (exposure to workplace hazards, pollution), natural or manmade air, water, soil pollutants, (NCI, 2015), ultraviolet rays, and radiation [40,41,42]. A plethora of studies have linked both prostate and breast cancer to sex hormones, suggesting that the two cancers might be connected to common pathogenic factors [6,11,18,41]. There are many risk factors that cause cancer, but due to the limitations of our research data, we cannot further analyze the possible underlying factors.

## 5. Conclusions

To the best of the researchers’ knowledge, the present study was the first to suggest that pharmacists have a lower all-cancer risk than the general population but higher prostate cancer risk (in men) and breast cancer risk (in women) compared to the general population. The study suggested that occupational hazards may play a role in occupational cancer, but other health-related (intrinsic and extrinsic) risk factors could also be investigated. Additional investigations are needed to clarify the risk and incidence of cancer in pharmacists in Taiwan. The findings from the study can help in guiding and developing appropriate strategic cancer prevention programs for pharmacists, both in Taiwan and around the world.

## 6. Limitations

This present study has some limitations. First, the study limited itself to an 11-year period (2000–2011). Some researchers (such as, Lee et al., 2015) have argued that this period may not be enough for a population-based cohort study; and based on that argument, additional studies including longer periods and more cases might be needed. Second, the study did not capture some pertinent information, such as the severity of an individual cancer type, the number of working hours and the extent or levels of risk or exposure to intrinsic and extrinsic factors. Besides, some data, such as personality traits, past history of cancer or family history of cancer, could not be accessed on the LHID 2000 data. Third, the study excluded study and comparison cohorts with a cancer history, which had implications on selection bias. Fourth, information on drinking alcohol, smoking, diet, lifestyle and family health history was unavailable to adjust for these potential confounders in data analyses. However, the impact from some of these factors might be minor because pharmacists are more likely to avoid unhealthy behaviors. Smoking and drinking alcohol is rare among pharmacists in Taiwan. The study results might not be generalizable to non-Chinese populations and populations with higher rates of obesity.

## Figures and Tables

**Figure 1 ijerph-18-12625-f001:**
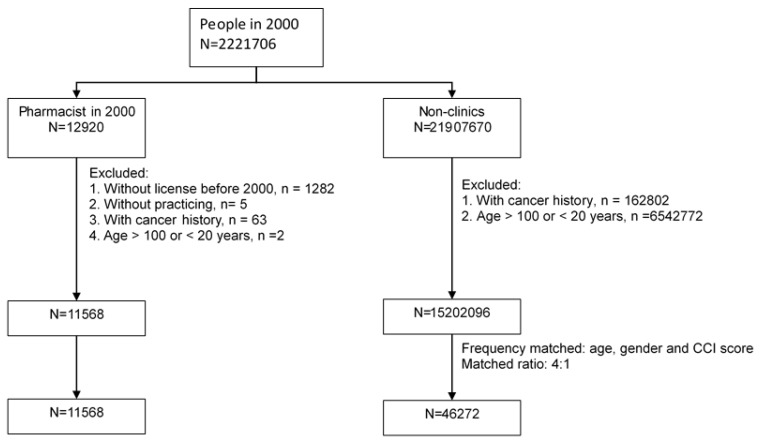
Flowchart showing the selection of the study population.

**Table 1 ijerph-18-12625-t001:** Distribution of age, gender and CCI score between pharmacists and comparisons.

	Pharmacist *N =* 11568		Comparison *N =* 46272	
	*n*	%	*n*	%
Men	4731	40.9	18924	40.9
Age, year				
20–34	5407	46.7	21,628	46.7
35–44	3576	30.9	14,304	30.9
45–54	2053	17.8	8212	17.8
55+	532	4.60	2128	4.60
Mean (SD)	37.4	(9.50)	37.4	(9.51)
CCI score				
0	11,420	98.7	45,680	98.7
1	112	0.97	448	0.97
2+	36	0.31	144	0.31
Work years				
1–2	3498	30.2		
3–5	4111	35.5		
5+	3959	34.2		
Workplace				
Medical center	1512	13.1		
Regional hospital	1887	16.3		
District Hospital	3220	20.1		
Clinics	3860	33.4		
Pharmacy	1989	17.2		

Note. General characteristics and demographics of the study cohort and comparison cohort. SD, standard deviation.

**Table 2 ijerph-18-12625-t002:** Cancer incidence and risk stratified by gender, age and CCI score.

	Pharmacist		Comparison					
	Event	Rate	Event	Rate	cHR (95% CI)	*p*-Value	aHR (95% CI)	*p*-Value
Overall	383	2.83	1547	2.96	0.95 (0.85–1.07)	0.40	0.96 (0.85–1.07)	0.43
Gender								
Women	224	2.80	714	2.29	1.22 (1.05–1.42)	0.01	1.23 (1.06–1.43)	0.007
Men	159	2.89	833	3.96	0.73 (0.61–0.86)	0.0002	0.73 (0.61–0.86)	0.0002
Age, year								
20–34	80	1.25	312	1.29	0.97 (0.76–1.24)	0.79	0.97 (0.76–1.24)	0.79
35–44	130	3.12	469	2.86	1.09 (0.89–1.32)	0.40	1.09 (0.90–1.32)	0.40
45–54	119	5.02	516	5.55	0.90 (0.74–1.10)	0.31	0.90 (0.74–1.10)	0.30
55+	54	9.37	250	11.20	0.83 (0.62–1.12)	0.23	0.83 (0.62–1.11)	0.21
CCI score								
0	370	277	1514	2.94	0.94 (0.84–1.05)	0.28	0.94 (0.84–1.06)	0.32
1	9	7.44	27	5.62	1.33 (0.62–2.82)	0.46	1.33 (0.62–2.82)	0.46
2+	4	11.32	6	4.56	2.55 (0.72–9.04)	0.15	2.16 (0.59–7.99)	0.25

Interaction p for cancer between pharmacist and gender was <0.0001; aHR: adjusted hazard ratio; cHR: crude hazard ratio; 95% CI: 95% confidence interval.

**Table 3 ijerph-18-12625-t003:** Cancer incidence and risk in different locations by gender.

	Pharmacist		Comparison			
LOCATION	Event	Rate	Event	Rate	aHR (95% CI)	*p*-Value
All						
Liver	27	0.20	206	0.39	0.50 (0.33–0.74)	0.0007
Breast (women only)	116	1.45	271	0.87	1.68 (1.35–2.08)	<0.0001
Lung	20	0.15	125	0.24	0.61 (0.68–0.98)	0.04
Thyroid	23	0.17	63	0.12	1.41 (0.88–2.28)	0.16
Colon rectum	40	0.30	168	0.32	0.91 (0.65–1.29)	0.60
Prostate (men only)	14	0.25	24	0.11	2.18 (1.12–4.21)	0.02
Kidney	13	0.10	30	0.06	1.67 (0.87–3.20)	0.12
Nasopharynx	12	0.09	41	0.08	1.13 (0.59–2.15)	0.71
Stomach	12	0.09	57	0.11	0.80 (0.43–1.50)	0.49
Bladder	12	0.09	34	0.07	1.33 (0.69–2.57)	0.40
Other	94	0.70	528	1.01	0.69 (0.55–0.86)	0.0008
Women						
Liver	5	0.06	25	0.08	0.77 (0.30–2.02)	0.60
Breast	116	1.45	271	0.87	1.68 (1.35–2.08)	<0.0001
Lung	9	0.11	48	0.15	0.73 (0.36–1.49)	0.39
Thyroid	20	0.25	47	0.15	1.65 (0.98–2.79)	0.06
Colon rectum	15	0.19	60	0.19	0.98 (0.56–1.72)	0.94
Prostate						NA
Kidney	4	0.05	6	0.02	2.62 (0.74–9.27)	0.14
Nasopharynx	1	0.01	14	0.04	0.28 (0.04–2.13)	0.22
Stomach	5	0.06	18	0.06	1.09 (0.40–2.93)	0.87
Bladder	2	0.02	5	0.02	1.56 (0.30–8.03)	0.60
Other	47	0.59	220	0.71	0.84 (0.61–1.15)	0.27
Men						
Liver	22	0.40	181	0.86	0.46 (0.30–0.72)	0.0006
Breast						NA
Lung	11	0.20	77	0.37	0.54 (0.29–1.02)	0.06
Thyroid	3	0.05	16	0.08	0.71 (0.21–2.44)	0.59
Colon rectum	25	0.45	108	0.51	0.88 (0.57–1.36)	0.55
Prostate	14	0.25	24	0.11	2.18 (1.12–4.21)	0.02
Kidney	9	0.16	24	0.11	1.43 (0.66–3.07)	0.36
Nasopharynx	11	0.20	27	0.13	1.56 (0.77–3.15)	0.21
Stomach	7	0.13	39	0.13	0.68 (0.30–1.51)	0.34
Bladder	10	0.18	29	0.14	1.30 (0.63–2.67)	0.47
Other	47	0.85	308	1.46	0.58 (0.43–0.79)	0.0005

aHR: adjusted hazard ratio; 95%CI: 95% confidence interval.

**Table 4 ijerph-18-12625-t004:** The association between cancer incidence and pharmacist characteristics by gender.

	All		Women		Men	
	aHR (95% CI)	*p*-Value	aHR (95% CI)	*p*-Value	aHR (95% CI)	*p*-Value
Age, year	1.07 (1.06–1.08)	<0.0001	1.06 (1.05–1.08)	<0.0001	1.08 (1.06–1.09)	<0.0001
Women vs. men	1.59 (1.27–2.00)	<0.0001				
CCI score	1.42 (1.02–1.98)	0.038	1.28 (0.67–2.47)	0.456	1.42 (0.96–2.09)	0.079
Work years						
1–2	1.04 (0.78–1.39)	0.772	1.15 (0.81–1.62)	0.436	ref.	
3–5	1.03 (0.81–1.32)	0.808	Ref.		1.29 (0.83–2.02)	0.257
5+	Ref.		1.03 (0.75–1.43)	0.842	1.14 (0.72–1.83)	0.576
Workplace						
Medical center	1.39 (0.89–2.16)	0.149	1.37 (0.84–2.23)	0.204	1.25 (0.42–3.71)	0.694
Regional hospital	Ref.		Ref.		Ref.	
District Hospital	1.31 (0.88–1.95)	0.191	1.12 (0.70–1.80)	0.636	1.99 (0.87–4.56)	0.104
Clinics	1.28 (0.88–1.88)	0.198	1.18 (0.76–1.83)	0.471	1.88 (0.83–4.23)	0.128
Pharmacy	1.59 (1.07–2.37)	0.023	1.40 (0.82–2.38)	0.215	2.14 (0.97–4.68)	0.058

aHR: adjusted hazard ratio; 95%CI: 95% confidence interval.

## Data Availability

We accessed the database of the Taiwan National Health Research Institutes. We are not eligible to duplicate and disseminate the database. For further access to the database, please contact the Ministry of Health and Welfare (Email: stcarolwu@mohw.gov.tw) for assistance. Taiwan Ministry of Health and Welfare Address: No.488, Sec. 6, Zhongxiao E. Rd., Nangang Dist., Taipei City 115, Taiwan (R.O.C.). Phone: +886-2-8590-6848.

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
