# Peer review of "Population-Based Study on Cancer Incidence in Pharmacist: A Cohort Study in Taiwan"

_ijerph, 2021, doi:10.3390/ijerph182312625_

Round 1
Reviewer 1 Report
The Cancer Incidence in pharmacists is addressed efficiently in this article and emphasized prostate (in men) and breast (in women) cancer risks in Taiwan. Since occupational hazards in the workplace are unavoidable, but safety training for pharmacists can help to reduce the rate of cancer risk in Taiwan and around the world. Hence, a suggestion for the author is to design a questionnaire-based investigation to assess the knowledge of pharmacists about occupational hazards and risk factors.
Comment 1)
In line 324, recheck the sentence” pharmacists are more likely than the general population of the general population to….”.
Comment 2)
In line 150, recheck the sentence “empirical analysis and final the final result because…..”
Comment 3)
What is the difference between workplace environment mentioned in line 116 & workplace in line 117.
Comment 4)
In 4. Discussion, the paragraph starting from line 310 to 316.
The author mentioned the difference in anticipated result and their findings. Give the proper justification for this difference.
Author Response
Dear Editor:
The authors thank the reviewers for their helpful and thoughtful comments and suggestions. In the document below we have listed our responses point by point. Reviewers’ comments are numbered and in Cambria font. Responses follow in Calibri font.
Best Regard.
Ming-Hung Lin

Reviewer 2 Report
See the attached document, please.

Author Response
Dear Editor:
The authors thank the reviewers for their helpful and thoughtful comments and suggestions. In the document below we have listed our responses point by point. Reviewers’ comments are numbered and in Cambria font. Responses follow in Calibri font.
Best Regard
Ming-Hung Lin

Reviewer 3 Report
The article by Lin Y W et al titled as “Population-based Study on Cancer Incidence in Pharmacist: a Propensity Score Matched Cohort Study in Taiwan” is a statistical analysis based on the data available from the government-database (Ministry of Health and Welfare) in Taiwan to reveal the pharmacist’s risk associated with cancer. The study is well designed, and the findings are interpreted appropriately. Also, the manuscript is well written. The article can be accepted for the publication; however, the authors should clarify –
Why did the authors specifically focus the study on data of “2000-2011”? Why are the authors, not including the recent data too to cover a longer period of time since it’s a population-based study?
Minor points-
In line 68: sentence “…which me that spread….” needs to be corrected.
In line 150: “...the final and final result…” should be corrected accordingly.
Paragraph (lines 354-360) can be rephrased for a better clarity.
Author Response

(The authors gave the same response as above.)

Round 2
Reviewer 2 Report
Please, see the attached document.

Author Response
Dear editor:
The authors thank the reviewers for their helpful and thoughtful comments and suggestions. In the document below we have listed our responses point by point. Reviewers’ comments are numbered and in Cambria font. Responses follow in Calibri font.
Thank you for your consideration. I look forward to hearing from you.
Sincerely,
Ming-Hung Lin
